# Long-Term Ship Position Prediction Using Automatic Identification System (AIS) Data and End-to-End Deep Learning

**DOI:** 10.3390/s21217169

**Published:** 2021-10-28

**Authors:** Kunihiro Hamada, Yujiro Wada, Jota Nanao, Daisuke Watanabe, Takahiro Majima

**Affiliations:** 1Graduate School of Engineering, Hiroshima University, Hiroshima 739-8527, Japan; d190785@hiroshima-u.ac.jp; 2Graduate School of Advanced Science and Engineering, Hiroshima University, Hiroshima 739-8527, Japan; waday@hiroshima-u.ac.jp; 3Kawasaki Kisen Kaisha, Ltd., Tokyo 100-8540, Japan; nanao.jota@jp.kline.com; 4Marubeni Corporation, Tokyo 100-8088, Japan; watanabe-daisuke@marubeni.com; 5Knowledge and Data System Department, National Maritime Research Institute, Tokyo 181-0004, Japan; majy@m.mpat.go.jp

**Keywords:** deep learning, AIS, ship position prediction, long-term, end-to-end

## Abstract

The establishment of maritime safety and security is an important concern. Ship position prediction for maritime situational awareness (MSA), as a critical aspect of maritime safety and security, requires a longer time interval than collision avoidance and maritime traffic monitoring. However, previous studies focused mainly on shorter time-interval predictions ranging from 30 min to 10 h. A longer time-interval ship position prediction is required not only for MSA, but also for efficient allocation of ships by shipping companies in accordance with global freight demand. This study used an end-to-end tracking method that inputs the previous position of a vessel to a trained deep learning model to predict its next position with an average 24-h interval. An AIS dataset with a long-time-interval distribution in a nine-year timespan for capesize bulk carriers worldwide was used. In the first experiment, a deep learning model of the Indian Ocean was examined. Subsequently, the model performance was compared for six different oceans and six primary maritime chokepoints to investigate the influence of each area. In the third experiment, a sample location within the Malacca Strait area was selected, and the number of ships was counted daily. The results indicate that the ship position can be predicted accurately with an average time interval of 24 h using deep learning systems with AIS data.

## 1. Introduction

Maritime transportation is recognized for its central role in the global supply chain considering it accounts for 90% of international trade by volume and 70% by value [1]. The United Nations predicted that the total volume of seaborne trade worldwide would increase by 3.2% from 2019 to 2022 [2]. Therefore, establishing safety and security in maritime transportation is essential. Maritime situational awareness (MSA) is a critical aspect of maritime safety and security that can be achieved through tracking, surveillance, and position prediction of ships [3]. Once a prediction of the ship position is obtained, decision-making and action planning can be supported at different information levels. However, ship position prediction for MSA requires a longer time interval than other tasks, such as collision avoidance and maritime traffic monitoring, which mainly use a short-term prediction from a high-precision real-time forecast spanning less than one hour [4]. Accordingly, we considered a prediction with a time interval ∆t longer than 12 h as a long-term prediction, and a prediction with a time interval between short- and long-term threshold as a medium-term prediction [5]. Studies on the long-term prediction of vessel position remain scarce despite its considerable potential for maritime applications, while almost all previous studies have focused on either near real-time predictions (short-term prediction) or predictions with a time interval lesser than 12 h (medium-term predictions). Long-term ship position prediction is required not only for MSA, but also for efficient allocation of ships by shipping companies in accordance with global freight demand. It can be utilized to monitor and assist the fleet, specifically, when communication with a ship operator breaks down owing to poor weather conditions or when a ship is in distress. It can also be implemented by shipping insurance and maritime investigators for investigation purposes. The ability to predict the long-term position of a fleet could prove not merely necessary, but vital for strategy formulation in the fast-changing dynamics of maritime industries.

Maximizing the potential of maritime big data is essential for predicting vessel positions. The automatic identification system (AIS) is a self-reporting message system on board a vessel that records its position and condition [6]. Each record of the AIS message contains the static and voyage-related information of the vessel and its dynamic information such as longitude, latitude, speed over ground, course over ground, and heading. As of July 2008, large vessels serving international routes were regulated to be outfitted with an AIS Class A device by the International Maritime Organization (IMO) [7]. Prior to the widespread use of maritime big data (i.e., AIS data), studies on vessel position prediction were conducted using data from radar or laser sensors, such as in the work of Perera et al. (2012) [8]. Since 2015, studies in the field have started using AIS as a historical data source for ship position information. Czapiewska and Sadowski (2015) conducted position prediction experiments using linear and nonlinear motion functions for location data compression of AIS records [9]. The high volume of AIS data accumulated over the years is a potential asset that needs to be explored, especially in the current age of artificial intelligence. With this big data of vessel position records, it is possible to predict vessel position by applying advanced machine learning (ML) techniques.

The objective of this study is to achieve a 24 h interval ship position prediction for MSA. Previous studies focused mainly on short- and medium-term prediction, ranging from 30 min to 10 h intervals as the AIS data were dense (closely packed between short time intervals) but in a limited period or timespan. Naturally, a vessel position prediction with long time intervals (e.g., 24 h) requires data with a long timespan, given that state-of-the-art ML algorithms such as deep learning require large amounts of data to make accurate predictions for longer intervals [10]. Accordingly, nine-year AIS data for capesize bulk carriers worldwide and deep learning (DL) were used to accomplish long-term prediction. End-to-end learning refers to training a learning system represented by a single deep network model that supersedes the preprocessing stages typically present in traditional pipeline designs; with extensive fleet data and computing power, the model can learn robustness to noise and generalization to data variation. First, the model performance was demonstrated for the Indian Ocean, followed by a comparison of the performance for six different oceans and six primary maritime chokepoints to investigate the influence of each area. Finally, the model performance for a sample area within the Malacca Strait was analyzed to simulate a practical application. The utilization of nine years of AIS data for capesize bulk carriers worldwide and the development of a generalized DL model from a dataset with a long-time-interval distribution for the long-term prediction of ship positions constitute the novelty of this study.

## 2. Related Work

### 2.1. Vessel Position Prediction Studies

The methods for forecasting vessel positions can be classified into three categories: trajectory-based, point-based, and motion-based methods. Overall, studies concerning ship position prediction are summarized in Table 1; only studies focused on predicting vessel position are listed. The results of one method are not equivalent to those of another method, because each method is specifically developed for the desired objective and the data used.

In the trajectory-based prediction method, clusters of vessel trajectories must be initially created and classified based on shipping routes from historical voyage patterns in the examined area, at times focusing solely on particular ships [11]. A ship position prediction can then be derived based on a formulation or extrapolation of these clustered trajectories. Because entire trajectories (or possibly port-to-port trajectories) of vessels in an observed area are required, this method has been implemented mainly for large areas (e.g., open waters, oceans, and straits). Several studies have developed unique route extraction algorithms to be used before the vessel position is predicted by deriving from its routes, such as in the work of Pallotta et al. (2014), Tu et al. (2020), and Mazzarella et al. (2015) [12,13,14]. We considered ∆t of Mazarella et al. (2016) to be effective for only 10 h because they merely demonstrated their trajectory-based algorithm on a particular target and route, where the 10 h threshold exhibited the most significant performance compared to those of the other algorithms. Dalsnes et al. (2018) conducted predictions using a Gaussian mixture model combined with the neighbor course distribution method [15]. Using k-nearest neighbor (kNN), Virjonen et al. (2018) compared the trajectories of a new ship and historical ships within the Gulf of Finland [16].

In the point-based method, the examined area is first transformed into non-overlapping cells or grids. The position of a ship inside the area is converted into a grid reference and defined by the occupied cells. The movement of these occupied cells indicates the movement of ships after a given time. A trained ML model then calculates the prediction of the next position of the ship. Duca et al. (2017) employed a kNN classifier for ship route prediction [17]. Kim et al. (2018) employed a combined hierarchical architecture of a convolutional neural network (CNN) for feature extraction with five separate fully connected neural networks (NNs), to predict the number of ships around the Korean port Yeosu up to 50 min ahead with initial AIS points reconstructed by interpolation [18]. Zhou et al. (2020) used three different architectures, including CNN, long short-term memory (LSTM), and bi-directional LSTM (bLSTM) integrated with CNN, to predict the inflow and outflow of gridded Singapore waters [19].

In the motion-based method, the subsequent vessel position is estimated using either motion functions or a trained ML model where historical information is used as input. Most researchers have used this method for short- and medium-term predictions within a small area (e.g., near ports, waterways, and restricted water areas) and particular trajectories. Juraszek et al. (2020) used an extended Kalman filter for fast short-term prediction in the distributed processing system Apache Flink [21]. Mao et al. (2017) generated ship position predictions 20 and 40 min ahead for the west coast of the US using an extreme learning machine (ELM) [20]. Duca et al. (2017) employed a kNN classifier to forecast ship movements around Malta [17]. Using an NN, Simsir and Ertugrul (2009) and Zhou et al. (2019) conducted predictions with intervals of 3 and 20 min, respectively, targeting vessels in narrow waterways [22,25]. Meanwhile, Zissis et al. (2016) attempted predictions in Greek waters up to 4 h ahead, focusing on the regular sea trips of passenger ships around the Aegean Islands [24]. Borkowski (2017) combined AIS location data and sensor data onboard a vessel to predict its subsequent position, aiming at collision avoidance [23]. Gao et al. (2018) used a bidirectional recurrent neural network (bRNN) with an LSTM unit seeking short-term predictions [26], and another study by Gao et al. (2021) combined the TPNet framework and LSTM for multi-step ship trajectory prediction based on four types of navigation stages [27].

### 2.2. Characteristics of Trajectory-Based and Motion-Based Methods

As long-term vessel position predictions have not been performed yet, determining a suitable method is vital for achieving long-term prediction. Consequently, the observed location (and dataset) must be considerably larger than the ship distance intervals to accommodate and capture their long-term movements; a small observation area limits long-term movement. The point-based method inevitably involves computing the entirety of cell information in each computation cycle, and thus the observed area cannot be too large because the computational costs would be exorbitant. Some researchers used CNNs by converting vessel positions from AIS data into a gridded ocean floor model as the input. Meanwhile, utilization of the trajectory-based method is expected to result in high accuracy for any time-interval threshold and area size; however, this method is not sufficiently flexible for adaptation to factors other than the observed variable. It also involves arduous work on route clustering and classification and trajectory reconstructions, in addition to anomaly detection owing to the partially incomplete and noisy AIS data. Furthermore, although we can create a database for millions of historical trajectories, ship trajectories are not fixed but quickly adapt to the weather and other conditions; ship routes evolve owing to various factors, such as climate change and the emergence of new ports and new routes [28]. In contrast, the motion-based method is relatively more flexible, efficient, and practical than the other methods, and, notably, can be effectively generalized for data variation. Table 2 shows the main differences between trajectory-based and motion-based methods for vessel position prediction.

Therefore, from the perspective of long-term position prediction on large international open waters without the consent of any restricted trajectory, a more general and updatable model with generalization to data variation and robustness to noise needs to be constructed that incorporates a fast prediction generation for predicting the long-term position of a vessel. In this study, a motion-based method was used to develop a generalized deep learning model for long-term vessel position prediction to overcome these challenges.

## 3. Data Exploration and Characteristic of This Study

### 3.1. AIS Data

We processed AIS data from exactEarth [29], considering that it was established as an AIS-data-service company that started deploying AIS receivers on satellites soon after the IMO regulation was implemented. As exactEarth offers the untapped potential of extensive AIS data of SOLAS-compliant ships, we took advantage of the capacity provided by deep learning systems to learn from large datasets to improve the overall performance.

The dataset comprises nine years of archived AIS messages of global capesize bulk carriers, from July 2010 to December 2018. This type of vessel exhibits proper records with less irregularity compared to those of smaller vessel types and more diverse shipping routes than containers and tankers [30]. The dataset alone is composed of over 3.5 million archived AIS messages from 1698 different IMO numbers (vessels).

Daily archived AIS messages have random time intervals ∆*t* between less than 1 h and up to 48 h, averaging 24-h intervals; the time-interval distribution is depicted in Figure 1. The fitted probability density function (PDF) of the overall dataset showed a standard Gaussian distribution. The data proliferated from 2010 to 2014 when satellite-based AIS became a standard practice. Although the dataset has an uneven time-interval distribution, it is considered adequate for long-term prediction because it contains years of extensive fleet data worldwide.

### 3.2. Ship Data

Ship specification data from IHS SeaWeb’s fleet data [31] were utilized to organize and validate the capesize bulk carrier vessels from billions of raw exactEarth data records. This dataset was also used to rectify the static information in each AIS message.

All datasets were collected independently from each source and stored in a single SQL relational database for easy access and management. An IMO number identifies every vessel, each of which is related to another piece of information in the dynamic and static data. When a model needs to be generated, a specified chunk of the AIS dataset is retrieved from the relational database and subsequently processed into a DL model. All scripts from preprocessing to postprocessing were written in Python.

### 3.3. Characteristic of Data for Long-Term Prediction and Originality of This Paper

In previous studies, such as in Zhang et al. (2020), trajectories with missing records over one day and lasting less than one day were excluded from the training and testing process [28]. However, in this study, we do not execute this pre-processing because eliminating the routes or trajectories with missing data can be applied only when they have been defined, that is, only when using other methods (see Table 2). Because of the nature of our dataset, we consider an AIS message to be an outlier when its time interval from the previous messages is more than 48 h. These outliers (individual AIS messages, not the entire trajectories) are removed from the training process but are retained in the testing process.

An end-to-end DL model was developed to supersede a vessel position prediction model that requires arduous preprocessing steps (e.g., trajectory reconstruction) as the time intervals of the AIS data vary considerably. A trajectory reconstruction becomes impractical as the time intervals and their variation grow. The asynchronous nature of AIS messages with time intervals of less than 3 min allows the application of a trajectory reconstruction from a short-interval to a long-interval dataset without a significant error. This is because the AIS points can be aligned using a simple method of sampling the nearest point in the time dimension that satisfies the *x* interval condition as $\lim_{\Delta t\to x}|\Delta t-x|=0$. Nonetheless, this preprocessing destroys information regarding latent variables [32]. Specifically, it causes distortion (the trajectory becomes compressed or simplified), on turn causing the acceleration (derived from speed over ground) and rate of turn to become unusable [33]. Then, if the values of the time interval ∆*t* are relatively large with uneven distribution, the interpolation causes a considerable error in the aligned positions [34].

In this study, the dataset has ∆t values ranging between almost zero and 48 h (in the absence of outliers), indicating that although the dataset consists of daily AIS messages (averaging 24 h), their exact time interval can be longer than 24 h (see Figure 1). The large gap in time intervals between two consecutive AIS messages signifies that the interpolation to create an evenly spaced dataset time series would result in large errors. For instance, in a time window of 24 h, a moving ship generally has an average distance interval of more than 400 km. Consequently, interpolating the ship position in this extended time window is expected to result in a significant error in the aligned positions, thus accumulating additional errors when predicting the subsequent position. Figure 2 illustrates the uncertainty of trajectory reconstruction of the long-time-interval data. The aligned data are prone to result in a large error; therefore, any application or method that requires trajectory reconstruction is considered impractical.

Based on the nature of the AIS dataset while achieving the objective (which eventually focuses on the long-time interval data and large observation area), performing a trajectory reconstruction is unattainable. Therefore, in this study, DL with the regular feed forward network (MLP) was employed to solve the long-term position prediction. Previous studies have also attempted to use this basic architecture but have been limited to shallow networks with limited time intervals and area sizes. We aimed for a larger-scale and longer timespan of the AIS dataset, and more importantly, developed a more general and updatable model from a dataset with a long-time-interval distribution, which has not been achieved thus far. By leveraging the advantages of deep learning systems supported by extensive fleet data and computing power, the model was developed to learn robustness to noise from extensive data with unrestricted trajectories in international open waters and to realize generalization to data variation of time intervals, vessel activities, statuses, and locations.

## 4. Methodology

### 4.1. Overall Prediction Methodology

A straightforward method that takes advantage of the capacity provided by deep learning systems with sufficient data and computing power is proposed to produce long-term ship position prediction. Fast result generation and generalization from the trained model make the proposed method suitable for practical use. The flowchart in Figure 3 summarizes the overall method for long-term vessel position prediction. Given a selected location and time range, a sizeable chunk of the AIS dataset is queried from the database. Next, a set of input features $X_{i}$ and a set of target features $y_{i}$ comprising vessel positions at the next time step t+1 are created. Each set is standardized and then input for training into two independent DL models with the same model properties. The model generates a prediction of the displacements in longitude and latitude directions that are uncorrelated with each other. Then, the results are passed through inverse standardization using the standardization parameters of the target features. Finally, the predicted ship position (i.e., latitude and longitude) is systematically evaluated and analyzed.

### 4.2. Input and Target Features

A set of AIS messages is first retrieved from the SQL database, given a boundary condition of location and time. The specified input and output features are then extracted from the dataset with vessel states at the current time step t as the input matrix $X_{i}$ and its location at the next time step t+1 as the target vector $y_{i}$. The input X is composed of 10 features: longitude λ, latitude φ, speed over ground $U_{og}$, course over ground $\psi_{og}$, heading $\psi_{h}$, time interval ∆t, distance d, Haversine distance (great-circle distance calculation between two points on a sphere given their coordinates) $d_{h}$, Manhattan distance (distance between two points measured along axes at right angles) $d_{m}$, and average speed $U_{avg}$. Meanwhile, the component of the target y follows the DL model output that targets particularly one of the two variables: the displacements of longitude ∆λ and latitude ∆φ between the current time step t and the next time step t+1. The input matrix and target vectors are expressed as follows:(1)$$X_{i}=\left\{\lambda_{t}^{i},\varphi_{t}^{i},U_{og}{}_{t}^{i},\psi_{og}{}_{t}^{i},\psi_{h}{}_{t}^{i},∆t_{t}^{i},d_{t}^{i},d_{h}{}_{t}^{i},d_{m}{}_{t}^{i},U_{avg}{}_{t}^{i}\right\}$$
(2)$$y_{i}{}_{\lambda }=\left\{∆\lambda_{t}^{i}\right\};\;y_{i}{}_{\varphi }=\left\{∆\varphi_{t}^{i}\right\}$$
where $i=\left\{0,1,\ldots ,m\right\}$, the subscript t represents the time step, $∆\lambda_{t}=\lambda_{t+1}-\lambda_{t}$ is the longitude interval, and $∆\varphi_{t}=\varphi_{t+1}-\varphi_{t}$ is the latitude interval.

Input features of λ, φ, $U_{og}$, $\psi_{og}$, and $\psi_{h}$ are extracted directly from the archived AIS messages, whereas some of the remaining inputs provide implicit information on the position history at the previous time step t−1. Further, the target timestamp (at the next time step t+1) is represented as a time interval ∆t and a rough distance to the target $d_{t}$. They are defined as follows:(3)$$∆t_{t}=T_{t+1}-T_{t}$$
(4)$$d_{t}=∆t_{t}{U_{og}}_{t}$$
(5)$$d_{h_{t}}=d_{h_{\left(\left(\varphi_{t-1},\lambda_{t-1}\right),\left(\varphi_{t},\lambda_{t}\right)\right)}}where\;\varphi ,\lambda \;in\;radians$$
(6)$$d_{m_{t}}=d_{h_{\left(\left(\varphi_{t-1},\lambda_{t-1}\right),\left(\varphi_{t-1},\lambda_{t}\right)\right)}}+d_{h_{\left(\left(\varphi_{t-1},\lambda_{t-1}\right),\left(\varphi_{t},\lambda_{t-1}\right)\right)}}$$
(7)$${U_{avg}}_{t}={d_{h}}_{t}/∆t_{t}$$
where T is the timestamp of the AIS message, and $T_{t+1}$ is referred to as the prediction timestamp. We utilized implicit information because it enables the DL model to generate predictions while using less information, rather than employing every possible explicit information directly from the AIS messages (the selection is described in Section 4.3.1). The above calculations (Equations (3)–(7)) are empirical formulas such that the extraction is performed at a high speed.

After retrieving the input and target, all input features $X_{i}$ are standardized, similar to the target feature standardization shown in Equation (8). The standard score of the target feature $y_{i}$ is calculated as follows:(8)$${y^{\prime }}_{i}=std\left(y_{i}\right)=\left(y_{i}-\bar{y_{i}}\right)/\sigma_{y_{i}}$$
where $y_{i}$ is the target feature containing $y_{i}{}_{\lambda }$ and $y_{i}{}_{\varphi }$, $\bar{y_{i}}$ is the mean of $y_{i}$, $\sigma_{y_{i}}$ is the standard deviation of $y_{i}$, and ${y^{\prime }}_{i}$ is the standard score of $y_{i}$. The standardization of the input and output features avoids any imbalance in the network parameters and loss calculations during training.

Two sets of standardized input and target features, $\left({X^{\prime }}_{i},{{y^{\prime }}_{i}}_{\lambda }\right)$ and $\left({X^{\prime }}_{i},{{y^{\prime }}_{i}}_{\varphi }\right)$, are then fed separately into two independent DL models with the same model properties.

### 4.3. Deep Learning Model

#### 4.3.1. Model Development

A DL model refers to the process of training and testing multilayered NNs that can learn complex structures and achieve high levels of abstraction [35]. Building a DL model to perform a specified task satisfactorily requires hours of iterative prototyping. The process is not a trivial task, even for a simple small model. Keras, an open-source high-level API framework capable of running on GPU clusters [36] was used to facilitate rapid model prototyping. The decision regarding the structure of the model and its input-target features was determined through our extensive experimentation and evaluation, focusing on the model objectives: performance and generalization on the observed areas.

A hold-out validation split was conducted to prevent information leakage from the test data. AIS data from July 2010 to December 2017 were randomly shuffled into a training set, and the dataset was disrupted with data from other vessels and timelines. Meanwhile, a dev set was randomly sampled from half of the 2018 data. The remainder was retained as a test set that had never been used for model prototyping. The split was designed to reflect the dev/test set as the recent AIS data. The dev set was used to validate the training set while having the same distribution as the test set. The randomly shuffled data combined with mini-batch training ensured rapid convergence of the gradient descent with minimal disturbances. Moreover, this combination prevented deep networks from conspiring to memorize the chronological sequences of vessel locations.

During the prototyping process, the most salient input features were selected by evaluating permutation feature importance (PFI) based on Fischer et al. (2018) [37]. After a model was trained and the model score s was computed, a corrupted matrix $X_{k,j}$ was generated by permuting feature j in the original matrix X. Subsequently, a new score $s_{k,j}$ was computed based on the prediction of the permuted data $X_{k,j}$. Finally, the PFI score $I_{j}$ was calculated as follows:(9)$$I_{j}=\frac{\frac{1}{K}\sum_{k=1}^{K}s_{k,j}-s}{s}\times 100\%$$

The permuted data $X_{k,j}$ were then returned to the original order. This step was repeated for all the features of the input data.

The PFI evaluation was conducted using the dev set at the prototyping stage to determine the extent to which the model relied on each input feature to generate predictions [38]. All irrelevant and insignificant features with minimum PFI scores were removed, and subsequently, a new feature was extracted and re-evaluated until the final set of features was established. Based on our evaluation, any timestep-based information added to the final inputs would not improve the model performance. Instead, it would overfit the model.

#### 4.3.2. Model Structure

The model was constructed as a DL model to solve regression problems. Compared with the classification problem that treats portions of the covered area as classes, regression problems can achieve a higher location resolution and, therefore, higher prediction accuracy [27]. In this investigation, the target features are the displacements of longitude ∆λ and latitude ∆φ directions. We performed the regression task by setting the target as a scalar value for analysis and objective evaluation in the prototyping process at the expense of running it twice.

The proposed model uses a deep network with six layers: five hidden layers with a sigmoid activation function {64, 64, 32, 32, 16} and a one-unit linear output layer. The mean absolute error (MAE) was employed as the loss function since it is less sensitive to outliers than the root mean square error (RMSE), and mini-batch gradient descent with a mini-batch size of 64 was used for training. We used an adaptive learning rate method involving the adaptive moment estimation optimization algorithm (Adam), an update to the RMSProp with momentum [39]. The learning rate was automatically reduced by a factor of 0.5 once the validation loss stagnated for 50 epochs. Moreover, to accommodate areas with low AIS data coverage (to avoid overfitting caused by deep networks), a validation-based early stopping was set to interrupt training once the validation stopped improving for 100 epochs. These in-training validation-based methods were solely used for the experiment and were not used in the prototyping stage [40].

### 4.4. Performance Evaluation Metrics

#### 4.4.1. Loss Score

Two separate models were trained for each standardized target feature ${y^{\prime }}_{i}$ of $∆\lambda_{t}$ and $∆\varphi_{t}$ values (scalar regression model). Subsequently, each value was discretely predicted. The predicted vessel position was then generated by inverse standardization using the standardization parameters of the target features as follows:(10)$${\hat{y}}_{i}=std^{-1}\left({{\hat{y}}^{\prime }}_{i}\right)$$
where ${\hat{y}}_{i}$ is the prediction vector after inverse normalization, and ${{\hat{y}}^{\prime }}_{i}$ is the prediction vector that combines the outputs of the two separate models (${{{\hat{y}}^{\prime }}_{i}}_{\lambda }$ and ${{{\hat{y}}^{\prime }}_{i}}_{\varphi }$). Both results (${\hat{y}}_{\lambda }$ and ${\hat{y}}_{\varphi }$) are then transformed into latitude ${\hat{\varphi }}_{t+1}$ and longitude ${\hat{\lambda }}_{t+1}$ by the addition of each component in the input (i.e., $\lambda_{t}$, $\varphi_{t}$). Finally, a new loss score for each result is calculated as follows:(11)$$MAE_{\lambda }=\frac{1}{m}\sum_{i=1}^{m}\left|{\hat{\lambda }}_{t+1}^{i}-\lambda_{t+1}^{i}\right|$$
(12)$$MAE_{\varphi }=\frac{1}{m}\sum_{i=1}^{m}\left|{\hat{\varphi }}_{t+1}^{i}-\varphi_{t+1}^{i}\right|$$
where ${\hat{\lambda }}_{t+1}^{i}=\left({\hat{y}}_{i}{}_{\lambda }+\lambda_{t}^{i}\right)$ is the prediction longitude, ${\hat{\varphi }}_{t+1}^{i}=\left({\hat{y}}_{i}{}_{\varphi }+\varphi_{t}^{i}\right)$ is the prediction latitude, $\lambda_{t+1}^{i}$ is the target longitude, and $\varphi_{t+1}^{i}$ is the target latitude.

#### 4.4.2. Metric Score

The metric scores for the combined results (i.e., location) were defined as the mean distance error (MDE) and mean angular error (MaE). The distance error is the haversine distance between the prediction and target (true position) in kilometers, signifying the extent to which the result (prediction) deviates from the true position (target). MDE can be a single evaluation metric for the long-term prediction of vessel location. However, in cases where the distance between previous positions and target positions is remarkably diverse (owing to time-interval variation), another evaluation metric is required.

The MaE as the second evaluation metric was calculated based on the three known points of the base, target, and predicted positions. The angular error between the target and the prediction is a spherical triangle that can be solved using the law of haversines [41]. Because the haversine distance between the prediction and the target is small, the formula is derived in combination with the spherical law of cosine. Accordingly, these metric scores were calculated as follows:(13)$$MDE=\frac{1}{m}\sum_{i=1}^{m}d_{h}{}_{\left(v,w\right)}^{i}$$
(14)$$MaE=\frac{1}{m}\sum_{i=1}^{m}C_{\left(u,v,w\right)}^{i}$$
(15)$$C_{\left(u,v,w\right)}=\arccos\left(\frac{cos\left(d_{h}{}_{\left(v,w\right)}\right)-cos\left(d_{h}{}_{\left(u,v\right)}\right)cos\left(d_{h}{}_{\left(u,w\right)}\right)}{sin\left(d_{h}{}_{\left(u,v\right)}\right)sin\left(d_{h}{}_{\left(u,w\right)}\right)}\right)$$
where u is the base point $\left(\varphi_{t},\lambda_{t}\right)$, v is the true position $\left(\varphi_{t+1},\lambda_{t+1}\right)$, and w is the predicted point $\left({\hat{\varphi }}_{t+1},{\hat{\lambda }}_{t+1}\right)$. The predicted position is in a vector space, and consequently, its distance error relative to the current position may not be proportional to its angular error. Figure 4 illustrates the relationship between distance and angular errors. Furthermore, the distance and angular errors can reflect the accuracy and precision of the prediction of the target, respectively.

## 5. Experimental Results and Discussions

### 5.1. Baseline Model

A non-ML baseline and a sanity check were used as model-building guidelines to conduct error analysis on bias and variance, gaining confidence in the overall performance of the DL models. The baseline replicates a human-level performance as a proxy for the Bayes optimal error in the classification problem [42].

The predicted positions were defined as the next timestep positions averaging 24-h intervals, and consequently, a conventional formula for the dead reckoning position (in seamanship, a position determined by plotting courses and speeds from a known position) of the ship could not be adopted as the baseline. The dead reckoning calculation is too simple for two distant points; while it is accurate in the Euclidian space, it fails to map the Earth as a great circle (also known as an orthodrome, is the largest circle that can be drawn around a sphere; Earth is not a perfect sphere, but as long as the Arctic and Antarctic Circles are not included, all meridians on Earth can be treated as great circles). An equation for the geodesic on a spherical surface, namely the great circle equation, is more accurate for planning routes [43].

Therefore, the geodesic calculation was established as a baseline model, computed based on the shortest distance on a spherical earth. Given a starting position ($\varphi_{t}$ and $\lambda_{t}$), current course $\psi_{h}{}_{t}$, and distance to the target (next position) $d_{t}$, the destination point along a (shortest distance) great circle arc is computed as follows:(16)$${\overset{ˇ}{\varphi }}_{t+1}=\mathrm{asin}\left(\sin\left(\varphi_{t}\right)\cos\left(\delta_{t}\right)+\cos\left(\varphi_{t}\right)\sin\left(\delta_{t}\right)\cos\left(\psi_{h}{}_{t}\right)\right)$$
(17)$${\overset{ˇ}{\lambda }}_{t+1}=\lambda_{t}+\mathrm{atan}2\left(\sin\left(\psi_{h}{}_{t}\right)\sin\left(\delta_{t}\right)\cos\left(\varphi_{t}\right),\cos\left(\delta_{t}\right)-\sin\left(\varphi_{t}\right)\sin\left({\overset{ˇ}{\varphi }}_{t+1}\right)\right)$$
where $\varphi_{t},\lambda_{t},\;\psi_{h}{}_{t}$ are in radians, $\delta_{t}=d_{t}/r_{earth}$ is the angular distance, and $r_{earth}$ is the earth diameter [44]. The results from Equations (16) and (17) were subjected to normalization to degrees.

The average distance interval of the ships was adopted as a sanity check, assuming that the next position always equals the present state. This approach is adopted to examine the performance evaluation metrics between the current position $\left(\varphi_{t},\lambda_{t}\right)$ and the next position $\left(\varphi_{t+1},\lambda_{t+1}\right)$.

### 5.2. First Experiment: DL Model for the Indian Ocean

#### 5.2.1. Experimental Setup

In the first experiment, a DL model was built for the Indian Ocean. The observed area is within 60° E–90° E longitude and 24° S–0° N latitude boundaries, covering 8.6 million square kilometers (calculated by geodesic area assuming the earth is a perfect sphere with a radius defined in WGS84 as 6,378,137 m). Compared to all previous studies, the boundaries were relatively larger, adequately capturing the continuous long-term movements of the vessels. Figure 5 presents the normalized density distributions of the dataset in the observed area from nine years of AIS messages involving 1531 different IMO numbers (vessels). This distribution was calculated using kernel density estimation (KDE), adopting a Gaussian kernel [45], in which the background geography was visualized using an open-source Python library Matplotlib Basemap toolkit for geospatial data processing [46].

The dataset was split into a training set and dev-test sets according to the hold-out validation split described in Section 4.3.1. The training set comprised more than 122,000 data records after omitting AIS messages with missing values and abnormalities, whereas each dev and test set contained more than 10,000 data records. The size of these sets was considered sufficiently large, thus providing high confidence in the model performance.

#### 5.2.2. Error and Sensitivity Analyses

The MAE, as the performance evaluation metric of the model, is shown in Table 3. These values correspond to the absolute deviation between the prediction point and the target point in degrees (longitude λ and latitude φ). The results from the geodesic calculations were computed based on the training set.

Interestingly, the DL model performance on the training set and dev set surpassed the geodesic calculation as the baseline model. The model displayed minimal bias, fitting the training and dev sets satisfactorily. In practice, the sole focus of a regression model is not the deviation between the training loss and dev loss (i.e., variance), but rather its prediction performance, that is, the dev loss. Nevertheless, the variance in the model was insignificant, indicating that the model generalized very well to the dev set without overfitting.

Moreover, hyperparameters sensitivity of the DL model was carried out while also applying regularization techniques to confirm further that the model generalized very well to the dev set. All hyperparameters combinations were permutated except the structure of hidden layers and their units to see the effect on prediction performance from hyperparameters’ influence. This permutation was repeated for other models with regularization: 20% and 40% dropout on all hidden layers and a model with batch normalization.

Figure 6 shows the hyperparameters sensitivity of all models. The selected hyperparameter proved to be the best combination for the standard DL model. The model with no regularization achieved the most optimum error compared to other models with regularization, confirming that the model was already generalized well without overfitting. Applying dropout worsened the model performances than applying batch normalization. Naturally, predictions of longitude values would have less accuracy than latitude values since the longitude values have a distribution twice as large as latitude. This natural divergence appears to negatively affect the model with batch normalization but at the expense of accuracy.

Based on these error and sensitivity analyses, we determined that the proposed deep learning model achieved an optimal error with generalization to the dev set and was ready for the evaluation on the unseen test set resembling a practical scenario.

#### 5.2.3. Results and Discussions

The performance evaluation metrics of the DL and baseline models are presented in Table 4. All results were computed using the test set. The average distance interval indicates the difference between the current vessel position and the subsequent position. Compared with these average values, the geodesic calculation appears to be reasonably accurate for long-term predictions.

The DL model adequately outperformed the baseline model and almost doubled its performance based on the metric scores. Given that the distance between the current position of a vessel and its position the next day was approximately 486 km, the geodesic calculation might be accurate for long route planning, yielding an average distance error of 40 km. However, the DL model generated more accurate and precise predictions with a slight mean distance error of approximately 25 km and a mean angular error of 1.8°. If we assume that the evaluation metric (i.e., MDE) of the average distance interval created an error of 100%, simulating a model performance similar to the classification problem, the geodesic calculation and DL model predicted the vessel positions with 8% and 5% error, respectively.

Moreover, several ensemble-based ML regressors have been investigated, including random forest, gradient boosted decision trees, and extreme gradient boosting algorithm (XGBoost) [47,48]. The hyperparameters were tuned using the exhaustive grid search method, and therefore all hyperparameters were optimized for each ML model in this manner. With the same input, output, and validation splits, the performances across all the models were broadly comparable. Nevertheless, despite performing more accurately than the geodesic calculation, their performances were not significantly different from each other, among which the DL model still achieved the best performance (13% better than the ML models).

### 5.3. Second Experiment: DL Models for the International Open Waters

#### 5.3.1. Experimental Setup

In the second experiment, 12 DL models were built for six open oceans and six maritime chokepoints with the same model properties to assess the influence of the respective areas. Open waters of the North Pacific Ocean (NPO), South Pacific Ocean (SPO), North Atlantic Ocean (NAO), South Atlantic Ocean (SAO), Indian Ocean (IO), and Philippine Sea (PS) were selected considering that vessels generally sail across these deep oceans without stopping. According to Rodrigue (2020), the primary maritime shipping chokepoints were selected to represent long-term vessel behavior near the ports and congested (high-traffic) waters, namely Gibraltar Strait (GS), South Africa Coast (SAC), Bab al-Mandab Strait (BMS), Strait of Hurmuz (HS), Laccadive Sea (LS), and Malacca Strait (MS) [49]. Altogether, the ocean and chokepoint areas correspond to the simple and complex movement of ships, respectively.

Figure 7 shows the location and size of each area (red rectangles). The size of the observed maritime chokepoint area was as large as that of the open ocean. Each has the same observation area size, within 30-interval longitude and 24-interval latitude, covering more than eight million square kilometers. The same procedure as in the first experiment was applied to the models: each used the same model properties and followed a similar hold-out validation split. We limited the worldwide observation to these 12 areas to avoid building an excessively large model. A large model is prone to fit noise from inferior-quality data and random vessel behavior contained in the massive AIS dataset. Regularization methods might be a constructive solution; however, they require an extremely large model to fit all nonlinear problems.

#### 5.3.2. Results and Discussions

The performance evaluation metrics of the models for each examined area are listed in Table 5. Again, the DL models substantially outperformed the geodesic calculation as the baseline model in all areas, especially in the chokepoint areas. The improvement score ($I_{s}$) was the only inference to determine the extent to which the DL model outperformed its counterpart baseline model based on the *MDE* scores. The improvement scores were notably higher in the chokepoint areas than in the open ocean areas, scoring more than a 50% improvement over the baseline model for almost all the chokepoint areas. The geodesic calculation appears accurate enough to predict moving vessels. However, it fails to predict vessel behavior near the ports and high-traffic areas (i.e., the chokepoint areas). With no information regarding vessel status, historical trajectory, or destination, conventional calculations would fail.

Both the DL model and geodesic calculation showed more accurate predictions for the ocean areas than the chokepoint areas, but not in the average distance intervals between the current position and the next position, indicating that more vessels have shorter distance intervals in the chokepoint areas. Almost all vessels were moving in the ocean areas, whereas in the chokepoint areas, many vessels were idle for berthing or resting at anchorages. These activities resulted in an extremely short distance interval between two consecutive AIS messages at any time interval. Many vessel activities have very low distance intervals in the chokepoint areas but not in the ocean areas. Consequently, the average distance intervals in the chokepoint areas would indicate lower performance evaluation metrics than the ocean areas. The irregularity of idle vessels and activities other than sailing with minimal distance intervals renders the *MaE* unsuitable as a metric score. The angle between the prediction and the target no longer corresponds to the accuracy of the prediction because the distance interval between two consecutive positions is extremely short. Accordingly, for the evaluation of the chokepoint areas, the *MDE* alone suffices as a single metric score.

#### 5.3.3. Discussions: Influence of the Area

Figure 8 presents the distance error distribution of the predicted positions based on the distance interval, where the distance errors are the haversine distance between the true position $\left(\varphi_{t+1},\lambda_{t+1}\right)$ and the predicted point $\left({\hat{\varphi }}_{t+1},{\hat{\lambda }}_{t+1}\right)$.

In the ocean areas, as most data have distance intervals between 400 and 600 km, the DL models generated continuous accurate predictions with a distance error below 40 km for 80% of the population over the entire test set. For the same 80% population, the geodesic calculation generated a distance error below 60 km with a more diverse distribution, where its standard deviation was twice as high as that of the DL models.

In the chokepoint areas, as most data have distance intervals distributed between 400 km and near zero, the DL models constantly generated accurate predictions with a distance error below 65 km for 80% of the population. In contrast, the geodesic calculation generated projections with distance errors below 100 km for only 65% of the population. Its projections resulted in grave errors when the distance intervals were broad, but the errors became less severe when the distance intervals were near zero. Overall, the geodesic calculation fails to make accurate projections for the complex movements of vessels in chokepoint areas.

Deep learning systems can learn from large datasets to effectively improve their performance to generate accurate predictions by discovering complex variations across all input features [50]. Accordingly, we computed the mean PFI score of the DL models on the ocean areas and chokepoint areas to determine which features play a vital role in improving the overall performance (shown in Figure 9). The scores were calculated based on the test set of each area and were grouped by ocean and chokepoint areas and then averaged. The input features were classified into five components: distance, speed, angle, time, and position.

Based on the results for both the chokepoint and ocean areas, the speed component of the vessel (i.e., $U_{og}$, $U_{avg}$) does not appear to be essential because the distance d already represents it, whereas the course over ground $\psi_{og}$ proves vital for making predictions.

Models in the ocean areas rely primarily on input features of angle and distance components but not on speed and position components. This is similar to the geodesic calculation, which generally proved accurate for position projections of vessels moving in a definite straight direction or making simple movements.

Meanwhile, models in the chokepoint areas rely more on the current position (i.e., $\lambda_{t}$, $\varphi_{t}$), implying that they can be used to develop high-level abstraction to alternatively predict the next position when vessels deviate from a simple trajectory. Furthermore, focusing on the distance component, they also rely more on the distance d as a rough estimate of the distance interval to the next position than the haversine distance $d_{h}$ and Manhattan distance $d_{m}$ as the distance interval between the current position and the previous position, suggesting that the previous position of the vessel at t−1 is not as important as its current position at t and a rough estimate projection at t+1. Nonetheless, the utilization of all input features may not suffice to cover the entire range of unexpected vessel behavior near ports or with complex movements.

The model performance can be enhanced by incorporating additional information regarding the vessel status, destination, or historical trajectory without a trajectory definition. Thus, this approach remains a topic to explore in our future work.

#### 5.3.4. Discussions: Influence of Data Size

The proportion of AIS messages varied widely according to the area location. This variability among different areas is a general characteristic of AIS data: rarely passed areas contributed only a limited amount of data, whereas congested areas contained a substantial quantity of information. Figure 10 presents the size distribution of the training and dev-test sets in each observed area based on the same hold-out validation split.

The proportion of the dev-test sets to the training set increased in the areas with the busiest shipping routes; for instance, the Malacca Strait (MS) area encompasses half of South East Asian waters, including the most congested maritime primary chokepoints and three secondary chokepoints in Indonesian waters [49]. The proportion was drastically reduced in the rarely transited areas, such as the Pacific Ocean. Consequently, areas with a small amount of test data would indicate lower confidence in the overall performance of the model than areas with extensive data. The K-fold validation split can solve this problem. Nonetheless, it would produce non-comparable models.

The dataset size affects deep learning performance, as a small amount of training data may degrade the performance [51]. We recreated the DL model for the Malacca Strait (the area with the most extensive dataset size) for each variation in the training set to show the effect of the data size on the model performance (see Figure 11). The most significant improvement in the performance occurred when the training set size exceeded 25,000. The model performance began to stabilize when the training set reached approximately 200,000. This effect may not represent all areas because three areas have a small dataset of less than 25,000 thresholds (i.e., NPO, SPO, BMS). However, all the models were trained with eight years of data and are thus still capable of producing accurate predictions.

### 5.4. Third Experiment: Application of the DL Model

#### 5.4.1. Experimental Setup

In the third experiment, a DL model was built for the Malacca Strait (MS) area, and a sample location was selected in which the number of ships was counted each day consecutively, resembling a simulation of practical application. In the MS area, ships traveled approximately 417 km daily on average (see Table 5). Thus, the Malacca Strait itself, between Malaysia and Indonesia (95.2° E–103.2° E longitude and 0.5° N–6° N latitude), was selected as the sample location, covering an area of half a million square kilometers. Figure 12 shows the sample locations in the MS area. As the highest density area, Singapore waters were not included in the sample because there would be many outliers in the northeast and southeast portions that enter the observed sample location. Nonetheless, long-term ship behavior near ports can be observed on Malaysia’s Teluk Rubiah port inside the location.

For the application, uninterrupted AIS data for a period of time were required as the test set. Consequently, the previous hold-out validation split cannot be applied because the test and dev sets are equivalently derived from shuffled 2018 data. In addition, the training and dev sets have to be generated in the same manner as the models in the previous experiment to produce a generalized DL model similar to the previous experiment.

Therefore, a different scheme for the hold-out validation split was developed. Randomly shuffled AIS data from July 2010 to September 2017 were used as the training set, while randomly shuffled data between October 2017 and September 2018 were used as the dev set. The last three-month data were not randomized and were retained chronologically as the test set. The predictions were processed sequentially, day by day, rather than instantaneously. Subsequently, the number of ships was counted at the selected locations every day.

#### 5.4.2. Results and Discussions

The fitted plots of the overall results from the DL models and geodesic calculations were compared to the actual data, as shown in Figure 13. The results are the daily prediction of the number of ships inside the Malacca Strait from October to December 2018. The r-value measures the linear relationship between the predictions and actual values. Compared to the actual data, a strong relationship with the r-value of 0.83 from the DL model confirms that predictions with an average time interval of 24 h using deep learning are possible, which is contrary to the belief that the motion-based method is not effective for long-term vessel location prediction. By using proper DL systems and massive data, this result confirms that an accurate long-term prediction of the vessel position can be generated with the straightforward motion-based method. In contrast, the geodesic calculation showed a weak relationship with an r-value of 0.51, suggesting that conventional calculations cannot be used for the long-term position projection of a vessel moving in congested waters or near the ports.

Moreover, Figure 14 shows the daily prediction of the number of ships inside the sample location for the entire month of November 2018. The estimated target values from the DL model and geodesic calculations were compared to the ground truth target values. The DL model proved that it can accurately predict the quantity inside the location for an entire day without additional information regarding historical navigational status. Meanwhile, as the geodesic calculation failed to predict the long-term position of a vessel moving in congested waters or near the ports, it also failed to predict the number of ships.

Detailed ship position predictions inside the sample location are shown in Figure 15. The ship position prediction from the DL model and geodesic calculation in the Malacca Strait was compared with the actual position on 10 and 11 November 2018. In this location, most ships were moving, and some were idle near Malaysia’s Teluk Rubiah port. Examining the actual data on 9 November, there was a line of ships moving from the northwest to the southwest, three of which were separated at the rear. These three ships moved to the southwest the next day, and another line of ships appeared from the northwest, which eventually was the start of a congested line of moving ships that appeared on the 11th. The DL model closely mimicked the position of these ships and the movement of the ground truth targets on 10 and 11 November, even for ships idling at the port. The number of ships and their positions were reliably predicted by the model. From these results, the dl model appears to have a sense of the dimension of the geographic coordinate system that can be or is often passed. In contrast, geodesic calculation failed to predict the position of the ships either in the moving position or idle near ports, and also resulted in a poor estimate of the number of ships inside the area.

## 6. Discussions on Robustness of End-to-End Deep Learning Models

### 6.1. Comparison of End-to-End DL Models

The MLP model shows a remarkable performance compared to the ensemble-based ML models and conventional approach from the dataset with the long-time-interval distribution. End-to-end means that the approach does not need a trajectory reconstruction and other preprocessing/postprocessing steps for making a prediction.

To further clarify the robustness of end-to-end DL-based models, we built two DL-based models: Deep RNN and LSTM. These sequence-based DL architectures were built with the same input, output, and validation splits to the MLP model so that their performances are broadly comparable (the structure was also constructed as closely as possible to the MLP model while also tuned to the most optimized). The input does not go through an interpolation or trajectory reconstruction to ensure an equivalent model. It already carries information regarding time interval ∆t and a rough distance $d_{t}$ to the target. The Deep RNN and LSTM models have five hidden recurrent layers with recurrent units of {64, 64, 32, 32, 16} and a one-unit linear output layer. The model was trained in mini-batch with the loss function of mean absolute error (MAE) optimized by Adam. The stacked deep recurrent layers were easily prone to overfit the training set (LSTMs are much easier), which was alleviated using a combination of recurrent dropout and recurrent layer normalization.

The Indian Ocean and Malacca Straits areas similar to the second experiment, representing the simple and complex movement of ships on the ocean and chokepoint areas, respectively, were selected as a definitive test for the DL-based models. The overall prediction methodology of these two recurrent models is similar to the second experiment. The models were also set to generalize on the data at which the training set was shuffled. The number of steps was considered one-time steps: the input only contains information at time step t (the implicit information of t−1 is included) to make a prediction at t+1, and thus this resembles a multivariate time series with a single time step.

The performance of the DL-based models is presented in Table 6. The results of the proposed MLP model in the ocean and chokepoint areas are slightly more accurate than the other DL architectures. However, with only a 1% improvement, the performance of the feed-forward model is considered similar to the recurrent models (RNN and LSTM) that have more trainable parameters. Applying more computational cost and complex models appears to negatively improve the performance. The MLP model took the fastest total training times on both areas with the least parameters than the other DL models. These results were obtained on a single Intel i7-9700K processor (8 CPU @3.60 GHz).

The patterns observed result from the small sequences (time steps), at which the regular feed-forward network is sufficient for this task. This is confirmed as the application of the LSTM model did not deliver better performance than the RNN model since LSTMs are much more capable to handle long sequential data than simple RNNs with limited short-term memory. The results correspond to those of Gers et al. (2002) who tested RNNs and LSTMs for time-series predictions, but the results were poor as simple multilayer perceptrons (MLPs) often outperformed LSTMs when applied to the same data [52].

### 6.2. Comparison of End-to-End DL Models on Small Dataset

A small dataset with short-time intervals near the ports was used to further analyze the robustness of the end-to-end DL models. The dataset was retrieved from the ATD2019 challenge dataset (https://gitlab.com/algorithms-for-threat-detection/2019/atd2019; accessed date 19 September 2021). All AIS training and challenge data with different properties were combined and used to construct a dataset with varying time-interval distribution. Figure 16a shows the short-time-interval distribution of the small dataset averaging around 10 s intervals. Most of the data have near-real-time intervals, and the rest vary up to 60 s. The dataset consisted of ships movements near ports located around north of Norfolk, Virginia, North America (see Figure 16b). In this dataset, course over ground $\psi_{og}$ is given but not heading $\psi_{h}$, and therefore, to accommodate the same 10 input features as the previous experiments, the heading was copied from the course over ground.

The dataset was split into a training set and dev-test sets according to the hold-out validation split with 80% data for training and the rest 20% for validation and test. The training set comprised more than 90,000 data records, whereas each dev and test set contained more than 10,000 data records. The size of these sets was considered sufficiently large, thus providing high confidence to the overall result.

Table 7 shows the results from the DL models and geodesic calculation on the small dataset. According to the second experiment results, the pattern of MaE on the small dataset confirms that the dataset consists of vessel behavior near ports with short-distance intervals such as berthing, idling, and maneuvering. The average distance interval between two consecutive AIS points is 24.6 m. At this small distance, the geodesic calculation is qualitatively similar to the dead reckoning calculation with an error of around 3.5 m. The DL models are 8% better than the geodesic calculation. Naturally, LSTMs can generate the most accurate short-term predictions of vessel position. However, on this dataset with the uneven distribution and without trajectory reconstruction, the MLP model generated predictions as accurate as those of the LSTM with almost half of the training times.

The results show that compared to the conventional approach, the DL-based models are not as effective as on the long-term prediction task, and the LSTM model delivers the same performance as the feed-forward networks since the time steps are limited (single time steps with 10 dimensions). This limited sequence allows the feed-forward networks to deliver optimum performance but would not maximize the potential of the sequence-based recurrent networks. Moreover, on the dataset with varying time-interval distribution and without trajectory reconstruction, the feed-forward networks generated predictions as accurate as those of the LSTM with faster training times and less trainable parameters.

## 7. Conclusions

This study employed DL models with experimentally defined properties using exactEarth AIS daily data. In the first experiment, a model for the Indian Ocean area was examined, and in the subsequent experiment, 12 models were investigated on open oceans and maritime chokepoints. In the last experiment, a selected sample location within the Malacca Strait area was examined, resembling a simulation of practical application.

The results demonstrated that predictions with an average time interval of 24 h were possible, confirming that the straightforward motion-based method can generate an accurate long-term prediction of the vessel position. The DL models generated more accurate predictions than the geodesic calculation as the baseline model in all areas. Predictions in the open ocean areas yielded higher accuracy than the chokepoint areas. However, compared to the geodesic calculation, the improvement scores were higher in the chokepoint areas than in the ocean areas because the geodesic calculation failed to predict vessel behavior near the ports and congested waters. The DL model can predict the complex movement of ships near ports and congested routes, whereas conventional calculations fail, with no information regarding vessel status, historical trajectory, or destination. The prediction accuracy was directly proportional to the amount of training data. Furthermore, the last experiment demonstrated that the DL model appears to have a sense of the dimension of the geographic coordinate system that can be or is often passed, wherein the chokepoint areas rely more on the input features of the current latitude and longitude of the vessels. Moreover, on the dataset with varying and uneven time-interval distribution and without a trajectory reconstruction, the proposed MLP model generated predictions as accurate as those of the LSTM with faster training times.

This paper is the first study to explore long-term ship position prediction using an AIS dataset with a long-time-interval distribution in a nine-year timespan for capesize bulk carriers worldwide, providing a valuable benchmark for future studies. The model properties may be enhanced by adding new essential input features regarding the vessel status, destination, or historical trajectory (or longer time steps) without a trajectory definition. In our future studies, we intend to improve the model performance by utilizing these techniques and incorporating other types of vessels, thereby increasing the complexity to the model. In addition, we will attempt a longer time interval for long-term prediction.

## Figures and Tables

**Figure 1 sensors-21-07169-f001:**
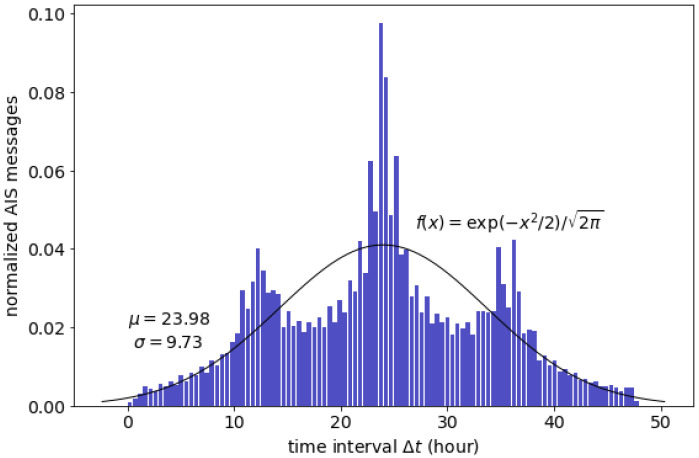
Time-interval distribution of the AIS dataset.

**Figure 2 sensors-21-07169-f002:**
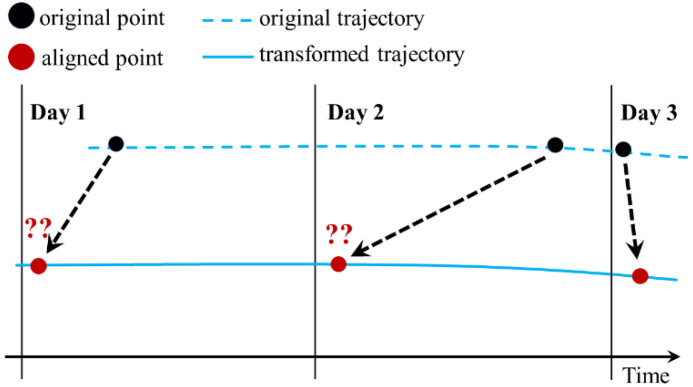
Uncertain trajectory reconstruction of long-time-interval data with uneven distribution.

**Figure 3 sensors-21-07169-f003:**
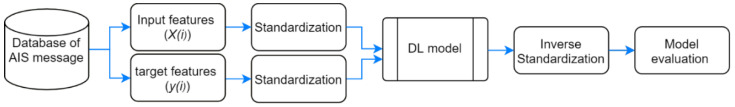
Flowchart of vessel prediction model with AIS and deep learning.

**Figure 4 sensors-21-07169-f004:**
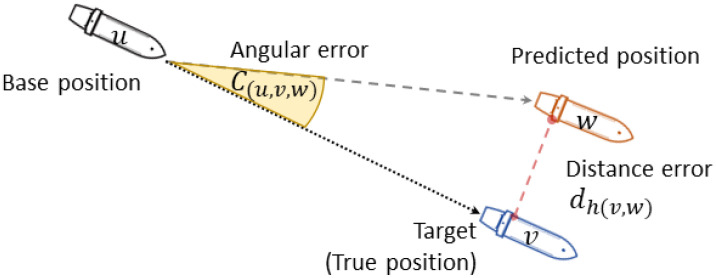
Relation between the distance error and the angular error.

**Figure 5 sensors-21-07169-f005:**
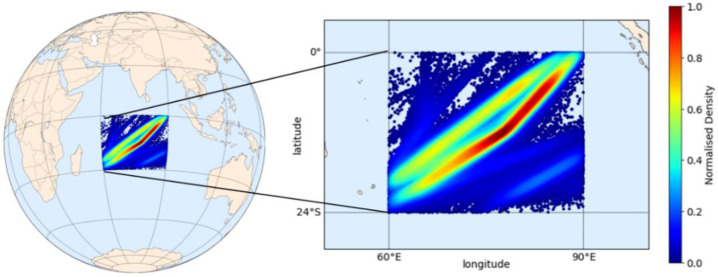
Normalized density distribution of AIS data in the observed Indian Ocean area.

**Figure 6 sensors-21-07169-f006:**
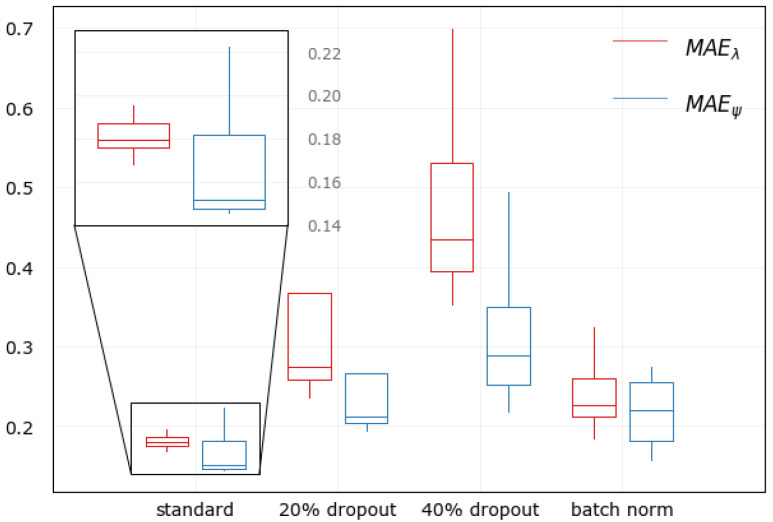
Sensitivity analysis of the DL model.

**Figure 7 sensors-21-07169-f007:**
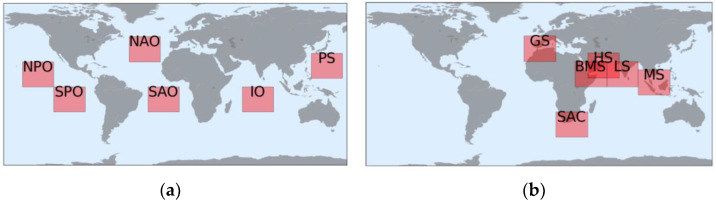
Location of the observed areas: (**a**) open oceans and (**b**) maritime chokepoints. NPO, North Pacific Ocean; SPO, South Pacific Ocean; NAO, North Atlantic Ocean; SAO, South Atlantic Ocean; IO, Indian Ocean; PS, Philippine Sea; GS, Gibraltar Strait; SAC, South Africa Coast; BMS, Bab al-Mandab Strait; HS, Strait of Hurmuz; LS, Laccadive Sea; MS, Malacca Strait.

**Figure 8 sensors-21-07169-f008:**
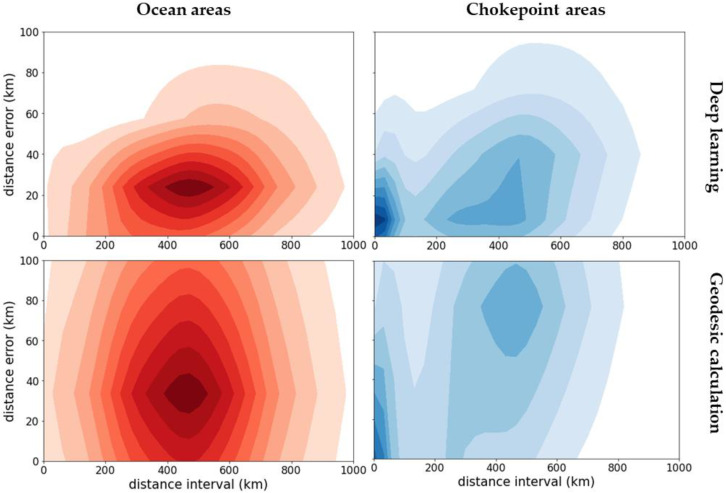
Distance error distribution of the predicted positions based on the distance interval on ocean areas (**left**) and chokepoint areas (**right**): DL model (**top**) and geodesic calculation (**below**).

**Figure 9 sensors-21-07169-f009:**
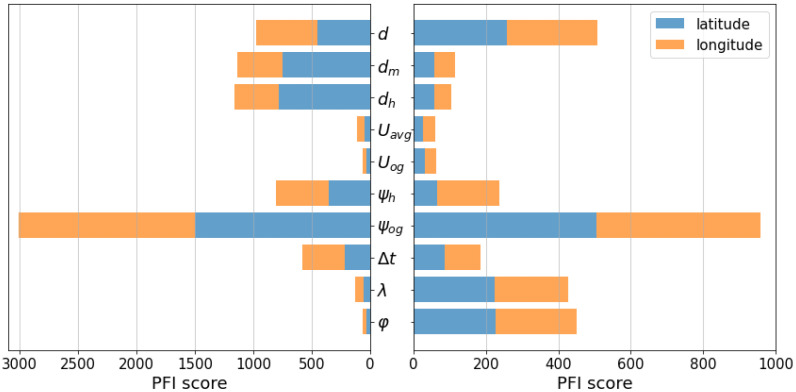
Mean PFI score of the DL models on the ocean areas (**left**) and chokepoint areas (**right**).

**Figure 10 sensors-21-07169-f010:**
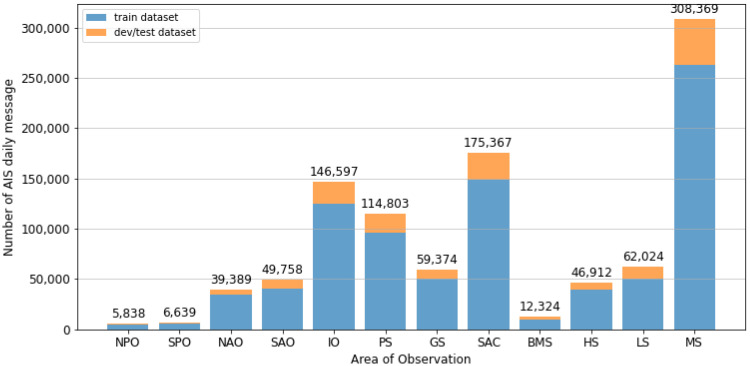
Size distribution of the training set and dev-test sets for each area.

**Figure 11 sensors-21-07169-f011:**
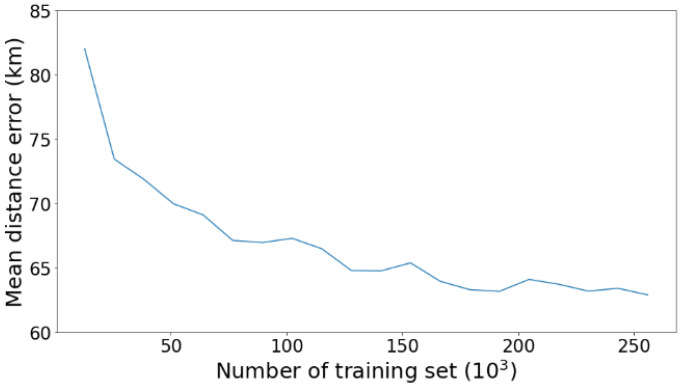
Effect of dataset size on the deep learning performance in Malacca Strait (MS).

**Figure 12 sensors-21-07169-f012:**
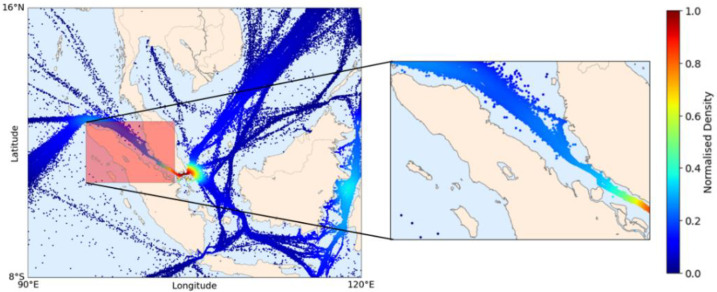
Location of the observation area (red-filled square): the Malacca Strait; the normalized density distribution is calculated based on the test set.

**Figure 13 sensors-21-07169-f013:**
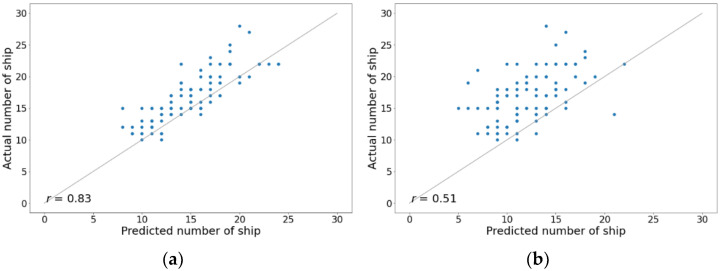
Fitted plots of the daily prediction of the number of ships in the Malacca Strait from October to December 2018: (**a**) DL model and (**b**) geodesic calculation.

**Figure 14 sensors-21-07169-f014:**
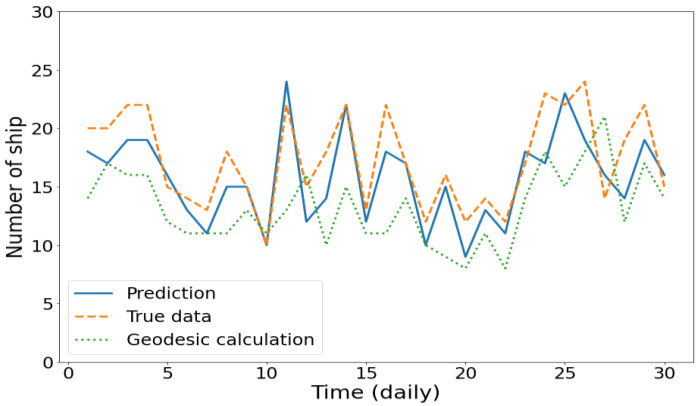
Daily prediction of the number of ships in the Malacca strait in November 2018.

**Figure 15 sensors-21-07169-f015:**
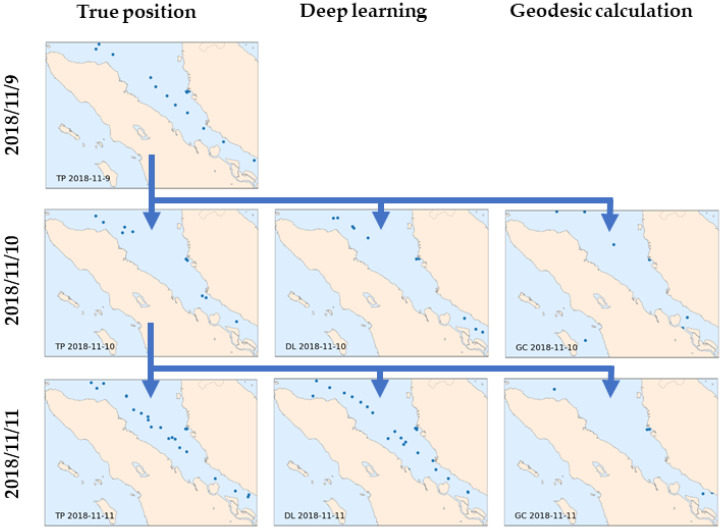
Ship positions in the Malacca strait on three consecutive days in November 2018: TP is the true position (**left**), DL is the deep learning prediction (**center**), and GC is the geodesic calculation (**right**).

**Figure 16 sensors-21-07169-f016:**
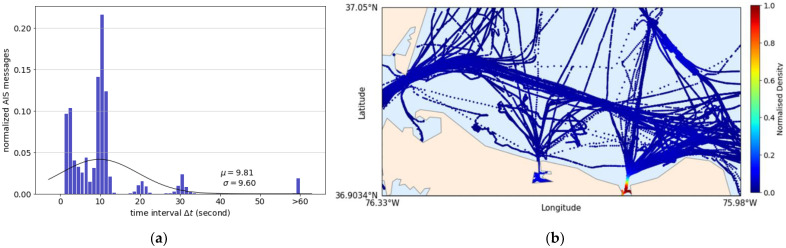
The small dataset properties: (**a**) Time-interval distribution; (**b**) Normalized density distribution.

**Table 1 sensors-21-07169-t001:** Vessel position prediction studies.

Prediction Method ^1^	Authors	∆*t* Threshold	∆*t* Prediction	Objective ^2^	AIS Data (Range)	ML Algorithm ^3^	Target Vessel ^4^	Target Area ^5^	Unrestricted Trajectory
T-b	[12]	Medium	8 h	MSA	5 months	-	Cg, Tg, Tk	OW	Yes
T-b	[14]	Medium	10 h	MSA	1 month	-	Cg	OW	No
T-b	[15]	Short	15 min	CA	1 year	-	*nr*	RW	Yes
T-b	[16]	Short	5 min	MTM	2 months	kNN	*nr*	RW	No
T-b	[13]	Medium	1 h	MSA	*nr*	ELM	*nr*	*nr*	Yes
P-b	[17]	Medium	1 h	MTM	1 month	kNN	Fs, Cg, Tk	RW	Yes
P-b	[18]	Short	50 min	MTM	2 years	CNN	Cg, Tk	RW	Yes
P-b	[19]	Short	5 min	MTM	1 month	CNN, LSTM	*nr*	RW	Yes
M-b	[8]	Short	<1 min	MTM	-	-	*nr*	RW	No
M-b	[9]	Short	5 min	DC	*nr*	-	Cg	RW	No
M-b	[20]	Short	40 min	AD, MSA	1 month	ELM	*nr*	OW	No
M-b	[21]	Short	8 min	MTM	3 months	-	*nr*	RW	Yes
M-b	[22]	Short	3 min	MTM	-	MLP	Tk	RW	No
M-b	[23]	Short	15 min	CA	*nr*	MLP	Fe	*nr*	No
M-b	[24]	Medium	4 h	MSA	*nr*	MLP	Ps	RW	No
M-b	[25]	Short	20 min	CA, MSA	*nr*	MLP	*nr*	RW	Yes
M-b	[26]	Short	10 min	CA	1 year	bLSTM	*nr*	RW	Yes
M-b	[27]	Short	<1 min	CA	*nr*	LSTM	Fe	RW	No
**M-b**	**Current study**	Long	**24 h**	**MSA, SA**	**9 years**	**MLP**	**BC**	**OW**	**Yes**

^1^ Prediction methods: T-b, trajectory-based; P-b, point-based; M-b, motion-based; ^2^ Prediction objectives: CA, collision avoidance; MSA, maritime situational awareness; SA., ship allocation; DC, data compression; MTM, maritime traffic monitoring; AD, anomaly detection; ^3^ ML algorithm or DL architecture used for vessel position prediction: SVM, support vector machine; ELM, extreme learning machine; kNN, k-nearest neighbors; MLP, multilayer perceptron; RNN, recurrent neural network; LSTM, long short-term memory; CNN, convolutional neural network; bLSTM, bidirectional long short-term memory; ^4^ Target vessel type: Cg, cargo, Tk, tanker, Tg, tugboat, BC, bulk carrier, Fs, fishing vessel, Fe, ferry, Ps, passenger ship; ^5^ Target area: OW, open water; RW, restricted water; *nr*, not reported.

**Table 2 sensors-21-07169-t002:** Primary differences between trajectory-based and motion-based methods for vessel position prediction.

	Trajectory-Based	Motion-Based
**Merits**	high accuracy for any time-interval threshold and area size	flexible, efficient, and can be generalized for data variation
**Demerits**	requires arduous work on pre-processing such as trajectory definition, classification, and reconstruction	requires machine learning where developing a model involves a CPU-intensive and specialized expertise

**Table 3 sensors-21-07169-t003:** Error analysis of the DL model.

Model	$MAE_{\lambda }$	$MAE_{\varphi }\;$
Geodesic calculation	0.225	0.226
Training set	0.163	0.145
Dev set	0.169	0.147

**Table 4 sensors-21-07169-t004:** Comparison of the model performance.

Methods	Loss		MDE (km)	MaE (Degree)
	$MAE_{\lambda }$	$MAE_{\varphi }$		
Average distance interval	3.553	2.564	486.1	-
Geodesic calculation	0.239	0.239	40.6	3.3
**Deep learning**	**0.174**	**0.149**	**27.0**	**1.7**
Random forest	0.197	0.174	31.3	2.1
Gradient boosted decision tree	0.198	0.171	31.4	2.3
XGBoost	0.197	0.175	31.6	2.2

**Table 5 sensors-21-07169-t005:** Comparison of the loss and metric scores of the DL model and baseline model on each observed area.

Area	DL Model				$I_{S}$ (%)	Geodesic Calculation				Number of Test Data	Average Distance Interval
	Loss		MDE (km)	MaE (deg)		Loss		MDE (km)	MaE (deg)		
	$MAE_{\lambda }$	$MAE_{\varphi }$				$MAE_{\lambda }$	$MAE_{\varphi }$				
NPO	0.18	0.13	**27**	1.8	**27**	0.20	0.23	37	3.1	471	485
SPO	0.13	0.11	**20**	1.5	**15**	0.13	0.15	24	2.1	499	484
NAO	0.30	0.22	**41**	3.7	**15**	0.33	0.28	48	4.5	2474	467
**SAO**	**0.13**	**0.11**	**19**	**1.7**	**24**	**0.15**	**0.15**	**26**	**2.5**	**4672**	**491**
IO	0.17	0.15	**27**	1.7	**33**	0.24	0.24	41	3.3	10,888	486
PS	0.25	0.31	**48**	8.0	**37**	0.48	0.46	77	10.5	9206	453
GS	0.54	0.27	**63**	21.7	**38**	0.78	0.53	102	20.2	4263	355
**SAC**	**0.35**	**0.23**	**46**	**15.4**	**50**	**0.59**	**0.54**	**92**	**15.0**	**13,337**	**389**
BMS	0.28	0.31	**51**	4.3	**55**	0.56	0.75	113	10.9	1251	473
HS	0.34	0.27	**52**	39.8	**58**	0.77	0.70	123	31.0	3734	259
LS	0.26	0.33	**51**	37.4	**50**	0.57	0.63	102	27.8	5975	273
MS	0.37	0.34	**63**	16.7	**60**	0.88	0.93	158	19.5	22,745	417

**Table 6 sensors-21-07169-t006:** Comparison of the deep learning model performance.

Areas	Methods	Loss		MDE (km)	MaE (Degree)	Training Time (s)
		$MAE_{\lambda }$	$MAE_{\varphi }$			
Indian Ocean (IO)	**MLP**	**0.174**	**0.149**	**27.0**	**1.7**	**1699**
	RNN	0.176	0.150	27.4	1.7	2758
	LSTM	0.177	0.148	27.3	1.7	1845
Malacca Strait (MS)	**MLP**	**0.376**	**0.337**	**62.8**	**16.9**	**4257**
	RNN	0.384	0.335	63.6	16.6	6561
	LSTM	0.381	0.336	63.4	16.2	6243

**Table 7 sensors-21-07169-t007:** Comparison of the model performance on the small dataset.

Methods	Loss		MDE (m)	MaE (Degree)	Training Time (s)	Total Params
	$MAE_{\lambda }(10^{-5})$	$MAE_{\varphi }(10^{-5})$				
Average distance interval	19.7	12.1	24.6	-	-	-
Geodesic calculation	2.4	2.1	3.5	29.2	-	-
**MLP**	**2.3**	**1.8**	**3.2**	**31.5**	**922**	**8545**
RNN	2.4	1.8	3.3	31.4	2537	19,041
**LSTM**	**2.3**	**1.7**	**3.2**	**31.4**	**1693**	**76,113**

## Data Availability

Some of the data presented in this study are openly available in reference number [29,31].

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
