# Peer review of "Long-Term Ship Position Prediction Using Automatic Identification System (AIS) Data and End-to-End Deep Learning"

_sensors, 2021, doi:10.3390/s21217169_

Round 1
Reviewer 1 Report
The paper is well structured and the language is satisfactory. The method is well detailed and very explanatory, and the use of several clearly described mathematical concepts confirm the method used. However, there are some inaccuracies and question I would like to ask the authors:
1- The author describes his work as better than others because the AIS data used does not suffer from the lack of data. However, it is not clear to me why the algorithm does not suffer from this lack of data. A possible solution could be that capesize bulk carriers doing standard routes and less dynamic than fishing vessels are easier to predict. Why did the author not eliminate the routes where the data is missing? Maybe the results would have been even better. Please improve this aspect.
2- In Pag 1, the author defines long term and medium term prediction, however a citation on this definition would be appropriate.
3- Pag 2, the author write "The objective of this study was to achieve" but in my opinion the present tense is better "is to achieve".
4 - The definition of end-to-end is unclear to me and should be improved in this context.
5- Tables and figures should be center aligned.
6- Repetition of "the location" in "Figure 5 shows the location and the location".
7- In Table 5, the better scores should be highlighted in bold.
In my opinion, the paper should be accepted after resolving these inaccuracies.
Reviewer 2 Report
The paper presents a deep learning based approach for long term ship position prediction using the AIS data. The underlying topic is very interesting and relevant to the journal. However, I find the methodological novelty to be very limited. Also, the comparison study is not sufficient to prove the superiority of the proposed approach. I suggest the authors to make following changes:
- Please consider including some other competing approaches. Geodesic approach is the most rudimentary approach for comparison. Try one of the motion model based approaches. Include a LSTM based approach. Using a sequence of timestamps to make a prediction is more important and relevant for real time monitoring. You should also look into Matlab Sensor toolbox. They have some implementations available for position prediction.
- Consider trying it on a smaller dataset with more challenges such as near the port, parking maneuver etc. For example: you should try your method on this challenge dataset (https://gitlab.com/algorithms-for-threat-detection/2019/atd2019) and report prediction performances.
- Change the value of the hyperparameters of your DL model and report the sensitivity and effects on prediction performance. Have you tried batch normalization or dropout?
- We do have lots of techniques available for short term prediction using the real time data. Can you elaborate what is the point of using long time projection?
